# Prediction of Response to Anti-Angiogenic Treatment for Advanced Colorectal Cancer Patients: From Biological Factors to Functional Imaging

**DOI:** 10.3390/cancers16071364

**Published:** 2024-03-30

**Authors:** Giuseppe Corrias, Eleonora Lai, Pina Ziranu, Stefano Mariani, Clelia Donisi, Nicole Liscia, Giorgio Saba, Andrea Pretta, Mara Persano, Daniela Fanni, Dario Spanu, Francesca Balconi, Francesco Loi, Simona Deidda, Angelo Restivo, Valeria Pusceddu, Marco Puzzoni, Cinzia Solinas, Elena Massa, Clelia Madeddu, Clara Gerosa, Luigi Zorcolo, Gavino Faa, Luca Saba, Mario Scartozzi

**Affiliations:** 1Department of Radiology, University of Cagliari, 09042 Cagliari, Italy; corriasgmd@gmail.com; 2Medical Oncology Unit, University Hospital and University of Cagliari, 09042 Cagliari, Italy; ele.lai87@gmail.com (E.L.); pi.ziranu@gmail.com (P.Z.); mariani.step@gmail.com (S.M.); c.donisi@studenti.unica.it (C.D.); sabagiorgio@live.it (G.S.); an.pretta@gmail.com (A.P.); marapersano@alice.it (M.P.); dario.spanu@gmail.com (D.S.); frabalconi@gmail.com (F.B.); francescoloi@hotmail.it (F.L.); valeria.pusce@gmail.com (V.P.); marcopuzzoni@gmail.com (M.P.); czsolinas@gmail.com (C.S.); dinetto13112012@gmail.com (E.M.); clelia_md@yahoo.it (C.M.); marioscartozzi@unica.it (M.S.); 3Department of Medical Oncology, IRCCS San Raffaele Scientific Institute, Vita-Salute San Raffaele University, 20132 Milan, Italy; nikina310788@gmail.com; 4Division of Pathology, Department of Medical Sciences and Public Health, AOU Cagliari, University of Cagliari, 09124 Cagliari, Italy; daniela.fanni@unica.it (D.F.); clara.gerosa@unica.it (C.G.); gavinofaa@gmail.com (G.F.); 5Colorectal Surgery Unit, A.O.U. Cagliari, Department of Surgical Science, University of Cagliari, 09042 Cagliari, Italy; simonadeidda86@gmail.com (S.D.); arestivo@unica.it (A.R.); gizorcolo@hotmail.com (L.Z.)

**Keywords:** metastatic colorectal cancer, angiogenesis, anti-angiogenic treatment, prediction of response, biologic factors, radiomics

## Abstract

**Simple Summary:**

In this review, we aim to analyze the molecular bases of angiogenesis in metastatic colorectal cancer (mCRC) and to provide an overview of potential predictive factors for treatment response to anti-angiogenic drugs in mCRC patients. We based our research on literature data focusing on circulating, tissue, and imaging biomarkers that can potentially be used to predict tumour response to anti-angiogenic inhibitors. In this perspective, biological factors and functional imaging seem very promising approaches and further research in these fields will provide clinicians with useful tools to improve patient selection and outcomes.

**Abstract:**

Colorectal cancer (CRC) is a leading tumor worldwide. In CRC, the angiogenic pathway plays a crucial role in cancer development and the process of metastasis. Thus, anti-angiogenic drugs represent a milestone for metastatic CRC (mCRC) treatment and lead to significant improvement of clinical outcomes. Nevertheless, not all patients respond to treatment and some develop resistance. Therefore, the identification of predictive factors able to predict response to angiogenesis pathway blockade is required in order to identify the best candidates to receive these agents. Unfortunately, no predictive biomarkers have been prospectively validated to date. Over the years, research has focused on biologic factors such as genetic polymorphisms, circulating biomarkers, circulating tumor cells (CTCs), circulating tumor DNA (ctDNA), and microRNA. Moreover, research efforts have evaluated the potential correlation of molecular biomarkers with imaging techniques used for tumor assessment as well as the application of imaging tools in clinical practice. In addition to functional imaging, radiomics, a relatively newer technique, shows real promise in the setting of correlating molecular medicine to radiological phenotypes.

## 1. Introduction

Colorectal cancer (CRC) is one of the most frequent malignant tumors, representing about 10% of cases of cancer across the globe [1]. Around 20% of patients with CRC present with metastatic spread at diagnosis, and 30% among those with early stage disease will develop metastasis during their life [2,3,4]. Thanks to the development and introduction into clinical practice of new therapeutic strategies, in the last ten years the median overall survival (OS) of metastatic colorectal cancer (mCRC) patients has reached 35 months; however, OS at five years still does not exceed 20% [5,6]. Nowadays, the standard of care for mCRC treatment is represented by first-line fluoropyrimidine-based chemotherapy, either single agent or in combination with oxaliplatin or irinotecan (doublet or triplet regimen), plus biological agents targeting either vascular endothelial growth factor (VEGF, bevacizumab) or epidermal growth factor receptor (EGFR, panitumumab or cetuximab). The treatment plan is chosen according to the mutational/molecular status of the tumor and the clinical features of the patient [7,8]. Immunotherapy with immune-checkpoint inhibitors (pembrolizumab) is restricted to mCRC patients with mismatch-repair deficiency (dMMR) or high microsatellite instability (MSI-H) [9]. The choice of second-line treatment is mainly based on the chemotherapy agent administered during the first line (switching to oxaliplatin if the patient received to irinotecan and vice versa) plus an anti-angiogenic drug (bevacizumab or aflibercept) unless contraindicated [4].

One of the main recent challenges in CRC patients is development of the ability to predict the evolution (which patients will develop metastatic disease) and response to treatment (which patients will have a good response).

mCRC is dependent on several molecular pathways; among them, the angiogenic signaling pathway represents a cornerstone for its development, growth, and spread. As a consequence, anti-angiogenic agents such as bevacizumab, ziv-aflibercept, ramucirumab, and regorafenib represent a crucial part of mCRC treatment [10,11,12,13,14]. Bevacizumab is a humanized monoclonal antibody targeting all VEGF-A isoforms; in phase III trials, it has been demonstrated to improve survival independently of the RAS/BRAF molecular profile when combined with chemotherapy in both the first- and second-line settings in bevacizumab-naïve and mCRC-pretreated patients [4,15,16,17]. Aflibercept is a decoy receptor consisting of a recombinant human fusion protein that prevents the interaction of VEGF-A, VEGF-B, and PlGF with their receptors. It is approved in the second-line setting in combination with irinotecan, 5-fluorouracil, and folinic acid (FOLFIRI) in mCRC patients progressing on oxaliplatin-based chemotherapy according to the results of the VELOUR trial [4,18]. Ramucirumab is a humanized monoclonal antibody anti-VEGF receptor 2 (VEGFR-2) that has demonstrated improved survival in combination with FOLFIRI in mCRC patients who previously received bevacizumab, oxaliplatin, and a fluoropyrimidine in the phase III RAISE study [4,19]. Regorafenib is a multi-tyrosine-kinase inhibitor (TKI) that, among its targets, inhibits the angiogenic receptors VEGFR-1,2,3; nowadays, it is a standard option in heavily pretreated patients on the basis of the phase III CORRECT trial [4,20].

However, a significant subset of patients does not respond to treatment due to gradual development of resistance or drug-related toxicity [21].

For these reasons, since the introduction of anti-angiogenic agents, there has been substantial interest in the identification of markers that help predict which subgroup of patients will benefit from angiogenic pathway inhibition [22].

In this review we analyze the molecular bases of angiogenesis, the use of circulating and tissue biomarkers, and advanced imaging approaches that can potentially be used to predict the tumor response to anti-angiogenic agents.

## 2. Angiogenesis Pathways

Angiogenesis represents a complex mechanism by which new blood vessels are produced starting from endothelial precursors. New vascular network growth is considered to be one of the hallmarks of cancer, and is regulated by the interaction between growth factors and their receptors [23]. In particular, this process is strictly dependent on a balanced equilibrium between pro- and anti-angiogenic factors as well as among multiple signaling pathways [24]. In the process of cancer growth, this equilibrium is interrupted and the pro-angiogenic factors prevail, leading to the “pro-angiogenic switch” in order to increase the supply of nutrients to cancer cells. (Figure 1). An important pro-angiogenic role in the VEGF family has been identified. This glycoprotein group includes VEGF-A, VEGF-B, VEGF-C, VEGF-D, and placental growth factor (PIGF) [25]. VEGF-E is a viral form (known as viral VEGF) encoded by the orf virus, whereas VEGF-F derives from snake venom [26,27]. These proteins bind to the extracellular immunoglobulin (Ig)-like domains of VEGF receptors (VEGFRs), which include VEFGR-1, VEGFR-2, and VEGFR-3 and belong to the receptor tyrosine kinase (RTK) family [28].

An increase in VEGF expression is typical for several types of cancer, including CRC. The causes of this increase can be both epigenetic, such as hypoxia, low pH, or inflammatory cytokines, and genetic, such as the activation of certain oncogenes (such as RAS, EGFR) or the inactivation of tumor-suppressor genes (such as p53 and many others) [29].

The binding of VEGF-A with VEGFR-2 is considered to be the most important mechanism in cancer angiogenesis, and leads to a cascade of different signaling pathways, mainly resulting in proliferation and migration of endothelial cells and promoting their survival and vascular permeability [30].

The VEGF-A gene consists of eight exons. Alternate 5′ and 3′ splice site selection in exons 6, 7, and 8 generate multiple isoforms. Over the years, several isoforms have been detected; among them, VEGFA165 has been identified as the most important in cancer angiogenesis [31]. It binds to VEGFR as well as to neuropilin co-receptors (NP1 and NP2), which seem promote cellular response to VEGF ligands [32,33].

The role of VEGFR-1 in cancer angiogenesis appears to be more complex, and is not yet fully understood. While this type of receptor binds VEGF with an affinity about ten times higher than VEGFR-2, the resulting signal transduction cascade is unclear; however, there is evidence of its role in angiogenesis [29,34].

The third receptor, VEGFR-3, does not bind VEGF-A; rather, it is involved in lymphangiogenesis by binding to VEGF-C [35].

Many other factors have a similar function to VEGFA in cancer angiogenesis, such as PIGF, fibroblast growth factor (FGF), hypoxia-inducible factor (HIF) -1α and HIF-2α, and platelet-derived growth factor (PDGF).

Therefore, cancer angiogenesis is a complex and heterogeneous mechanism. In CRC, VEGF-A remains the main identified factor and one of the main molecular targets (Figure 2) [28,29].

## 3. Biomarkers of Angiogenesis

The use of anti-angiogenic agents in mCRC is now a therapeutic standard of care in the first- and second-line therapy; however, there are no validated predictive biomarkers which allow the response to these drugs to be predicted. Currently, the evidence indicates some predictive value for particular circulating serum, tissue, and advanced quantitative imaging approaches.

### 3.1. Tissue-Based, Genetic Polymorphisms

Several single-nucleotide polymorphisms (SNPs) of the VEGF gene have been investigated in CRC and hepatocellular carcinoma [36,37,38,39,40,41,42].

The CC genotype of rs3025039 polymorphism of VEGF-Ac.*237C>T was significantly related to time-to-treatment failure (TTF) in 46 mCRC patients treated with bevacizumab. Lower OS and progression-free survival (PFS) were observed in patients harboring at least one T allele, whereas the VEGF-A rs699947 A/A and ICAM-1 rs1799969 G/A allele were associated with longer survival in mCRC patients [43,44,45,46].

Regarding CD133, patients with the CC genotype in rs3130 had a significantly lower OS with bevacizumab regimens for advanced mCRC [47]. As for Fms-related tyrosine kinase 1 (FLT1), higher PFS and OS were observed in mCRC patients with the rs9513070 AA genotype, while the rs9513070/rs9554320/rs9582036 GCA haplotype was significantly associated with shorter PFS and OS [48]. In RAS wild-type mCRC patients treated with a bevacizumab-based regimen, a significant association of MKNK1 SNP rs8602 with PFS was observed, and the AA genotype carriers had a shorter median PFS [49]. Moreover, in both univariate and multivariate analyses, a significant association of interleukin (IL) 8 polymorphism (c.-251TA+AA) with shorter PFS was observed in mCRC [50]. As for IL-6, advanced CRC patients treated with bevacizumab-based therapies and harboring the rs2069837 G allele had shorter PFS [51]. Finally, BMAL1 SNPs (rs7396943, rs7938307, rs2279287) were correlated with poorer PFS and OS, and high BMAL1 expression in tumor cells was correlated with unfavorable clinical outcomes in bevacizumab-treated CRC patients [52].

Therefore, these SNPs are a promising predictive marker for the management of the patients treated with bevacizumab. Giampieri et al. retrospectively analyzed the correlation of different VEGF-A, VEGF-C, and VEGFR-1,2,3 SNPs with PFS and OS in 138 mCRC patients treated with regorafenib. When angiogenesis genotyping was performed, only VEGF-A rs2010963 maintained an independent correlation with PFS and OS, meaning that it may be useful for helping to better select optimal candidates for regorafenib therapy [44].

### 3.2. Circulating Biomarkers

Circulating VEGF is the most studied potential predictive factor for anti-angiogenic treatment in mCRC patients. The majority of studies have focused on bevacizumab.

In research by Duda et al. into plasmatic VEGF, PIGF, and VEGFR-1, only VEGFR-1 was related to prediction of response and tolerability to bevacizumab [53].Martinetti et al. explored various circulating prognostic biomarkers (VEGF, PDGF, SDF-1, osteopontin, and CEA) in patients receiving bevacizumab in three randomized clinical trials. Higher levels of VEGF and SDF-1 were likely to be associated with worse prognosis, especially in terms of OS. Moreover, increasing CEA values during treatment resulted in significantly worse prognosis independent of disease extension [54].Two phase III studies (HORIZON II and III) evaluated baseline levels of VEGF and soluble VEGFR-2 (sVEGFR-2) as prognostic and predictive biomarkers. High baseline VEGF was associated with worse PFS in both studies and with worse OS in the HORIZON II study. However, these results were not uniformly confirmed, and further studies are necessary to clarify the role of circulating VEGF as a predictive marker [55].In the multicentric prospective randomized ITACa trial, VEGF-A, eNOS, EPHB4, COX2, and HIF-1α mRNA levels were evaluated at baseline and during therapy according to objective response (ORR), PFS, and OS. Reduction in eNOS and VEGF levels from baseline to the first clinical evaluation showed better OS than the others, and might suggest a response to bevacizumab for this category [56].Abajo et al. described higher baseline levels of epidermal growth factor (EGF) and macrophage-derived chemokine (MDC) along with lower levels of IL-10, 6, and 8 in treatment-respondent mCRC patients. Treatment exposure increased MDC and decreased IL-8 levels, suggesting that a set of inflammatory and angiogenesis-related serum markers might be associated with the efficacy of the bevacizumab-containing regimen [57].An exploratory preplanned analysis of serum pro-angiogenic factors (SErum aNgiogenesis-cenTRAL) was conducted in 72 mCRC patients among the participants of the phase II CENTRAL (ColorEctalavastiNTRiAlLdh) study in order to find potential serum biomarkers able to predict response to bevacizumab in combination with FOLFIRI in the first-line setting. In this study, the early increase of FGF-2 serum values was identified as a biomarker to ameliorate the selection of patients for this therapeutic strategy; indeed, patients experiencing an increase of FGF-2 during the 8-week timepoint from baseline had an improved median PFS (12.85 vs. 7.57 months) (HR: 0.73, 95%CI: 0.43–1.27, *p* = 0.23 and 12.98 vs. 8 months, HR: 0.78, 95%CI: 0.46–1.33, *p* = 0.35, respectively) [58].In a post hoc analysis of the VELOUR trial, VEGF-A and PIGF were significantly higher in patients pretreated with bevacizumab than in those who had not previously received anti-angiogenic agents. In patients randomized to the placebo arm, survival was shorter in case of higher levels of VEGF-A (>144 pg/mL) and PlGF (>8 pg/mL) at baseline (9.6 vs. 12.9 months and 9.7 vs. 11.7 months, respectively), suggesting that they might reflect acquired resistance to bevacizumab. Conversely, in the aflibercept group, improved OS and PFS were observed regardless of baseline VEGF-A or PlGF levels, confirming aflibercept activity even in patients with bevacizumab-induced resistance [18].Currently, the phase II biologically-enriched DISTINCTIVE study aims to prospectively validate VEGFR-2 plasma levels as predictive factor for the efficacy of aflibercept plus FOLFIRI in RAS wild-type mCRC patients progressing after first-line treatment with oxaliplatin, fluoropyrimidines, and anti-EGFR monoclonal antibodies [59].

### 3.3. Circulating Tumor Cells (CTCs) and Free Nucleic Acids

CTCs and free nucleic acid detection in peripheral blood represent a promising strategy for predicting response to treatment, including anti-angiogenic drugs [60].

Although the role of CTCs has not yet been clearly defined yet, and despite the limitations of the technology used for their detection, various studies are investigating their potential application in clinical practice in mCRC patients receiving anti-angiogenic agents [61].

In a study by Cohen et al., mCRC patients with high CTC count (≥3CTC/7.5 mL) showed worse PFS and OS; moreover, these CTC values were predictive of worse outcomes irrespective of the treatment received. Of note, almost 50% of patients had been treated with bevacizumab [62].Another clue to a potential predictive role in response to anti-angiogenic agents was provided by Rahbari et al., who demonstrated a correlation between CTC detection and circulating angiogenic factors as well as an association with lower levels of EGF and FGF [63].VISNU-1, an open-label multicenter randomized phase III trial, was performed in an mCRC population selected based on their baseline CTC count, with the aim of identifying those patients who might benefit most from bevacizumab plus FOLFOXIRI versus bevacizumab plus FOLFOX6. VISNU-1 considered CTC counts ≥ 3 as the cut-off for the high-risk mCRC patient population; thus, patients with at least three CTCs were randomly assigned to one of the two arms. FOLFOXIRI-bevacizumab improved PFS compared to FOLFOX-bevacizumab (12.4 months vs. 9.3 months, respectively; *p* = 0.0004). According to these results, bevacizumab-FOLFOXIRI might be considered an adequate treatment option for mCRC patients with ≥3 CTCs [64].

Circulating DNA (ctDNA) is becoming crucial from the perspective of precision medicine and in the selection of patients in CRC [65,66,67]. Thus, it has been investigated as a biomarker for anti-angiogenic treatment:Zhou et al. analyzed cell-free DNA from CRC patients; POLR1D amplification and expression influenced cell proliferation and induced VEGF upregulation, which has been suggested to be involved in bevacizumab resistance [68]. In another study, twenty-one mCRC patients treated with first-line bevacizumab plus chemotherapy underwent plasma collection at various timepoints; DNA was extracted and sequenced with a panel of 90 oncogenes. All patients harbored 1–6 trunk mutations in ctDNA and showed a range of 1–89% in the mutant allele frequency (MAF). MAF variations were significantly related to disease status (decreasing in case of disease remission and increasing in case of progression; *p* < 0.001). More specifically, improved survival was observed in patients who experienced MAF reduction to below 2% at remission [69].Another prospective study of CRC patients aimed to investigate the correlation between genetic polymorphism of VEGF-A with survival using the DNA extracted from peripheral blood. In a univariate analysis, there was a significant association (OR = 0.32; *p* = 0.048) between genotype CC of the VEGF-A -1498C>T polymorphism and the presence of CRC liver metastasis [70].An open-label study by Wong et al. simultaneously analyzed ctDNA in plasma samples and tumor biopsies on different days of treatment with regorafenib in mCRC patients [71]. ctDNA was inversely correlated with PFS, and the presence of KRAS mutations was associated with shorter PFS.In the RAISE study, Tabernero et al. identified VEGF-D as a potential predictive biomarker for the efficacy of anti-angiogenic treatment in the second-line setting [19].A retrospective analysis of the CORRECT study showed how DNA mutational status obtained from the plasma of mCRC patients might be a noninvasive tool to analyze tumor genotype while at the same time finding clinically relevant mutations other than those identified in tumor tissue. Moreover, in a subgroup of patients selected according to mutational status and protein levels, regorafenib seemed to lead to improved clinical outcomes [72].

### 3.4. MicroRNA (miRNA)

miRNAs include a class of small single-stranded noncoding RNAs able to regulate gene expression in the post-transcriptional phase. The dysregulation of various miRNAs is involved in CRC carcinogenesis signaling pathways [73]. Thanks to their stability and protection from RNase-mediated degradation, miRNAs can be identified in biological samples [74].

To date, even if tissue-specific miRNA signatures have been detected, further research is required to identify a set of CRC-expressed miRNAs that can be used in the screening phase for CRC and adenomas. However, the use of miRNAs for the management of CRC treatment is more desirable.

Because miRNAs have been shown to be involved in tumor angiogenesis for both pro-angiogenic and anti-angiogenic factors (e.g., receptor tyrosine kinase signaling protein, HIF, VEGF, and EGF), their potential role as predictive biomarkers is worthy of further investigation.

It is known that oncogenic miRNAs and tumor suppressor miRNAs have different levels of expression in the various stages of CRC, from initial development to disease progression. Furthermore, because miRNA regulation in CRC is stage-specific, the identification of miRNA sequences during disease progression might reflect prognosis and might be indicative of response to treatment. In this perspective, the let-7, miR-21, miR-29a, and miR-17-92 clusters seem to play an important role.

In mCRC, miRNA-126 has been extensively investigated and is a major player in angiogenesis regulation. More specifically, high expression leads to increased VEGF-A signaling in endothelial cells, making it a promising biomarker for anti-angiogenic therapies [75,76].

On the other hand, miR-181a and miR-181b have shown a promising prognostic role in CRC, as high levels of these biomarkers have been associated with poor survival [77].

In stage III CRC patients, neoadjuvant chemotherapy has been demonstrated to improve survival and is now the standard of care [78]. Conversely, in stage II surgery alone is an effective treatment approach, and for this reason the indiscriminate administration of adjuvant chemotherapy is controversial [79,80,81]. Almost 30% of these patients belong to a poor prognostic group who will develop disease recurrence and as such are considered at high risk. To date, factors that can help clinicians to identify who might benefit from adjuvant chemotherapy include, for example, extramural venous invasion (EMVI), grade of differentiation, and bowel obstruction/perforation at diagnosis. However, these biomarkers are not specific, and further research is required to investigate potential biomarkers in order to improve the selection of stage II patients who are more likely to benefit from adjuvant treatment and to avoid unnecessary toxicity in low risk patients.

Among potential prognostic biomarkers, upregulation of miR-21 is of particular interest [82,83]. High expression in tumor tissues of 29 CRC patients was related to nodal and distant metastatic spread; these findings were confirmed in a further 156 cases, where a correlation with EMVI, liver metastasis, and advanced stage was found [83,84]. Higher expression of miR-21, as well as of miR-93 and miR-103, was identified in a cohort of CRC patients with liver metastasis [85].

In stage II CRC, high levels of miR-29a were reported to be related to recurrence risk [86]. The biological explanation for this increased metastatic risk lies in the targeting of tumor suppressor KLF4, which induces matrix metalloproteinase-2 upregulation and E-cadherin downregulation [87].

Based on the above-mentioned preliminary results, more studies are needed to transfer the predictive role of miRNAs from research to clinical practice.

## 4. Advanced Quantitative Imaging Approaches in Disease Detection and Outcome Prediction

CRC patients’ management has been significantly ameliorated as a result of recent achievements in imaging assessments. Indeed, more information on tumor biology and predictive biomarkers for treatment efficacy can be suggested by innovative diagnostic methods, including but not limited to diffusion-weighted imaging (DWI), fluorodeoxyglucose positron emission tomography (FDG-PET), and dynamic contrast-enhanced magnetic resonance imaging (DCE-MRI).

### 4.1. EMVI

EMVI is represented by tumor cells which exceed the muscularis propria, and can be found in 17–52% of CRC cases. It is known to be related to increased local invasion, disease recurrence, and poorer survival outcomes. EMVI can be efficiently detected by magnetic resonance imaging (MRI), which allows for more precise determination of local staging and evaluation of vascular invasion, making it essential in treatment management for rectal cancer patients before radical surgery. Indeed, based on MRI-detected EMVI (mrEMVI) findings, clinicians may choose an intensified treatment for patients who are candidates for preoperative chemo-radiotherapy. A molecular biomarker for accurately predicting the presence of EMVI has been recently found in the CpG island methylator phenotype (CIMP) [88,89]. Other studies have recognized an association between the hyperexpression of CD34 in tumors and the presence of mrEMVI [90]. Another interesting biomarker contributing to angiogenesis and frequently expressed in human CRC is thymosin beta 4 (Tβ4) [91]. Tβ4 is preferentially expressed in tumor cells undergoing epithelial mesenchymal transition (EMT) at the invasive front of the tumor, facilitating CRC progression [92]. Moreover, a progressive decrease in the expression of E-cadherin on the cell surface in Tβ4-reactive tumor cells undergoing EMT was observed, where the loss of E-cadherin was related with the loss of cell–cell adhesion and the migration of isolated spindle tumor cells towards the blood and/or lymphatic vessels [93].

### 4.2. Radiomics

In recent years, the new imaging tool of radiomics has brought a new perspective to the analysis of tumors [94]. According to the principles of radiomics, images are composed of data, and harbor a huge amount of information requiring in-depth analysis. One of the main forms of information that can be obtained from a radiomic analysis has to do with tumor heterogeneity; different radiomic features can be used to obtain information about this feature, i.e., skewness, kurtosis, etc. [94]. Tumor heterogeneity was one of the first parameters to be correlated with treatment response in patients with mCRC. In a study by Joa Ahn et al. from 2016 involving 235 patients with liver metastasis of CRC analyzed with computed tomography (CT), lower skewness, higher mean attenuation, and narrower standard deviation (SD) were independently associated with response to chemotherapy [95].

Radiogenomics is another approach based on computational tools able to extract and analyze a huge number of quantitative variables from radiologic imaging regarding tumor heterogeneity and phenotype, for example, shape, intensity, and phenotype. The information is then related to the clinical factor, laboratory, gene expression, and proteomic data.

Currently, although limited and preliminary, some evidence of the application of radiomics and genomics in CRC is available. Huang et al. and Li et al. independently developed two radiomics models that can preoperatively predict lymph node metastasis [96,97]. Liang et al. found and developed a preoperative radiomics signature for discriminating stage I-II from stage III-IV CRC [98].

Research on radiogenomics aims to demonstrate the association of quantitative imaging parameters with peculiar patterns of gene expression. Various imaging tools have been explored to identify a correlation between image variables and the mutational status of K-ras, which notoriously is an independent prognostic and predictive factor for anti-EGFR agents in CRC. In a study by Shin et al., K-ras mutations were more frequent in polypoid cancers and in those with longer axial length or increased ratio of the axial length to the longitudinal. The most promising data on the identification of K-ras mutations derives from multiparametric imaging, which combines the evaluation of 18F FDG uptake, CT texture, and perfusion [99].

Regarding angiogenesis, van Griethuysen et al. proved that radiomics could predict treatment response with performance comparable to that of expert radiologists when using EMVI status (along with TN-stage, MRF/EMVI status, and size/signal/shape) as a prediction tool in rectal cancer patients [100]. Radiomics models predicted response similarly to the morphology prediction performed by expert radiologists (AUCs of 0.69–0.79 and 0.67–0.83, respectively). Nevertheless, no significant predictive role was confirmed for radiomics based on semi-automatically generated segmentations with no manual input [100]. Qu et al. evaluated a dynamic radiomics-based model for predicting the efficacy of anti-angiogenic agents in mCRC patients with liver metastasis in order to help physicians in diagnosis and treatment. The application of dynamic radiomics variables performed better in predicting anti-angiogenic treatment response than conventional radiomic variables, appearing as a promising predicting tool worth of further research [101].

In locally advanced rectal cancers, a novel radiomic signature proved to be more sensitive than clinical-radiological findings in predicting disease-free survival (DFS), and the combination of clinical–radiological features and radiomic features significantly improved the ability to estimate DFS (*p* =  0.001, 0.005 on training and validation sets, respectively) [102].

However, despite the flourishing of studies that focus on radiomics, a great deal must still be done in order to determine their actual correlation with the molecular basis before it can pass from bench to bedside as a valid clinical tool. The only deeply and clearly understood cellular/molecular explication of the correlation between radiomics and clinical evolution is tumor heterogeneity, which cannot account for all the correlations that have been found thus far between treatment response and radiomic features. Liver cancer is another field of translational research on radiomics; studies have focused on microscopic venous invasion (MVI), a poor prognostic factor not identifiable with classical imaging [103]. Segal et al. demonstrated a correlation between 91 genes of the “venous invasion signature” and CT scan identification of “internal arteries” and absence of “hypodense halos” [104]. The same imaging biomarkers in association with “tumor–liver difference” resulted in very accurate predictive factors for MVI, early recurrence, and poor prognosis in a study by Banerjee et al. [105].

Finally, one of the main problems of radiomics is its strong dependence on reader experience and inter-reader agreement. Thus, this limit needs to be broken in order to obtain more reproducible data, including the type of radiomics study. In this setting, artificial intelligence (AI) and deep learning algorithms could represent ground-breaking tools for overcoming these limitations [106].

However, additional prospective translational studies need to be performed in the context of colorectal cancer in order to correlate imaging, notably radiomic features, with molecular pathways.

### 4.3. Functional Imaging

Functional imaging allows for mapping the distribution of markers of tumor viability, such as the perfusion or metabolic activity of cancers.

FDG-PET is able to reveal metabolic activity of cancer cells after treatment and before the occurrence of morphological and dimensional changes. Thus, when performed serially, it might be a useful tool for tailoring treatment in the neoadjuvant setting, in particular in rectal cancer, where FDG-PET response is an independent prognostic and predictive factor in patients undergoing chemo-radiotherapy [107,108]. A prospective study by Lee et al. assessed the changes on 18F-FMISO and 18F-FDG PET scans as a measure of hypoxia and metabolism in 15 mCRC patients receiving anti-angiogenic agents and evaluated the correlation with clinical outcomes and circulating angiogenic biomarkers. The findings showed that while FMISO PET-assessed hypoxia had only minor changes after initial treatment with anti-angiogenic therapy, it was associated with therapeutic response. FDG PET uptake changes according to Standardized Uptake Value (SUV) max and total lesion glycolysis (TNG) were associated with response to anti-angiogenic therapy. Furthermore, circulating VEGF and osteopontin were significantly correlated with the FDG and FMISO PET parameters [109].

Despite increasing studies on whole-body DWI, the evidence to date is insufficient to support its use to replace PET/CT [110]. Because cell death and vessels changes occur earlier than dimensional modifications, DWI might become an easy-to use and early predictive tool of tumor response in the future [111].

CT (DCE-CT) or perfusion CT, which examines vessel variables, e.g., blood flow, blood volume, and mean transient time, has been investigated as a potential prognostic biomarker in rectal cancer as well. In particular, studies have found lower blood flow in the primary cancer in patients with metastatic growth, which is associated with worse survival outcomes [112,113]. As for its predictive role for chemo-radiation, data on DCE-CT are equivocal, as low baseline perfusion levels were related with poor response in a study by Bellomi et al., whereas according to research by Sahani et al. the results were the opposite [114,115].

DCE-MRI has been suggested by preliminary data as a functional imaging biomarker for patients treated with anti-angiogenic agents, as it can be used to assess the extravasation of paramagnetic contrast agents along with their uptake and signal intensity changes.

At the present time, the above-mentioned imaging tools cannot be applied in clinical practice; however, they represent a starting point and scientific basis for further research; thus, their application in routine clinical practice in the near future cannot be excluded.

## 5. Conclusions

Today, anti-angiogenic treatment represents a cornerstone of mCRC patient management in both the first- and second-line settings [3,4,5,6,7,8,9,10,11,12]. Unfortunately, the real OS gain from bevacizumab, aflibercept, and ramucirumab emerged from clinical trials and real-life data is about 1.4 to 1.6 months, which is quite marginal in the clinical history of a mCRC patient [3,4,5,6,7,8,9,10,11,12,16,17,116,117,118,119]. The same concept applies even more to the role of regorafenib in further lines of treatment [20,120]. Moreover, certain patients do not benefit from anti-angiogenic agents and are exposed to toxicities that might be avoided [21]. In the era of precision medicine, crucial efforts have been made over the years to individuate potential predictive factors in order to ameliorate the selection of mCRC candidate patients for anti-angiogenic treatment and to identify a reliable biomarker; unfortunately, this objective has not yet been reached. Research has focused especially on tissue-based genetic polymorphisms, circulating biomarkers, CTC and ctDNA, miRNAs, and more recently on imaging tools.

As is clear from the above, the search for predictive factors for anti-angiogenic drugs has been very active in the last twenty years, but without consistent results that can be applied in clinical practice to improve the selection of patients and ameliorate the therapeutic potential of these agents. This demonstrates the difficulty of identifying reliable biomarkers or imaging tools in this setting.

In conclusion, while the research scenario is evolving, it requires further larger prospective studies with the hope of finding (as certainty cannot be provided at present based on the parameters that have been explored to date) and validating a truly useful tool for predicting response to angiogenesis inhibitors and ameliorating the selection of mCRC patients leading to improved outcomes.

## Figures and Tables

**Figure 1 cancers-16-01364-f001:**
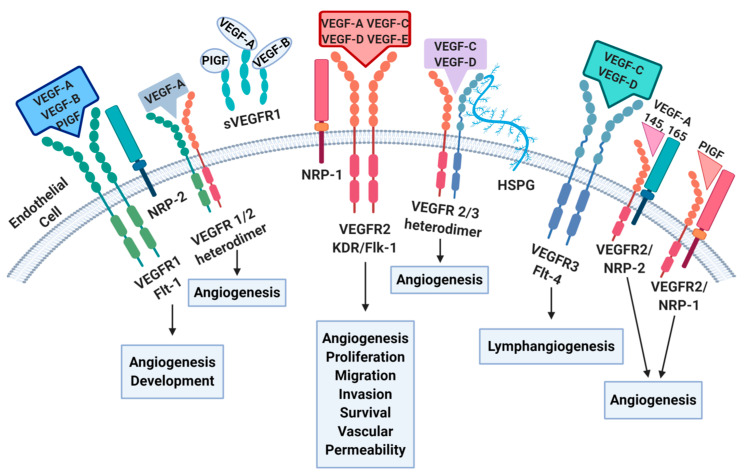
Angiogenesis ligands and receptors. Angiogenesis pathways lead to various biological consequences depending on the interaction between specific growth factors and their receptors. Abbreviations: Flt: Fms-related tyrosine kinase 1; HSPG: Heparan sulfate proteoglycans; KDR/Flk: Kinase insert domain receptor/Fetal Liver Kinase 1; NRP: neuropilin; PIGF: placentar growth factor; VEGF: vascular endothelial growth factor; VEGFR: vascular endothelial growth factor receptor.

**Figure 2 cancers-16-01364-f002:**
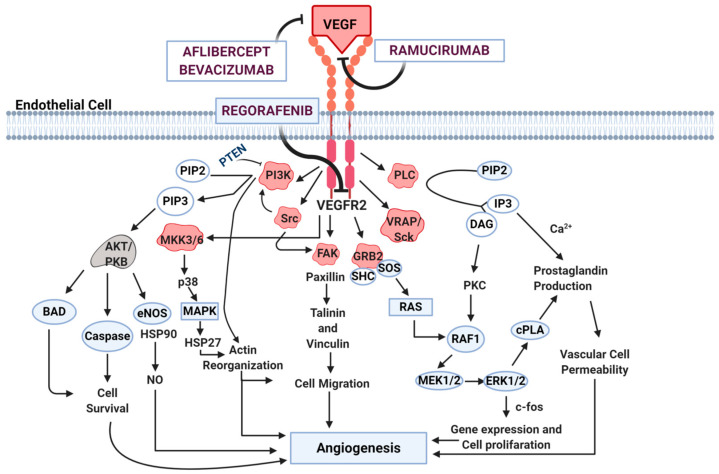
Angiogenesis pathway. The interaction between VEGF and VEGFR leads to the activation of downstream signaling which ultimately results in angiogenesis activation. Specific anti-angiogenic drugs such as bevacizumab, ramucirumab, aflibercept, and regorafenib block angiogenic cascade activation. Abbreviations: AKT/PKB: Protein Kinase B; BAD: BCL2 Associated Agonist Of Cell Death; Ca^2+^: Calcium ions; c-FOS: human homolog of the retroviral oncogene v-fos (FBJ murine osteosarcoma viral oncogene homolog); cPLA: Cytosolic phospholipase A; DAG: diacylglycerol; e-NOS: Endothelial-nitric oxide synthase 3; ERK: extracellular signal-regulated kinase; FAK: focal adhesion kinase; GRB2: Growth factor receptor-bound protein 2; HSP: heat shock proteins; IP3: Inositol trisphosphate; MAPK: mitogen-activated protein kinase; MEK: mitogen-activated protein kinase kinase; MKK3/6: mitogen-activated kinase kinases; PI3K: phosphatidylinositol 3-kinases; PIP: phosphatidylinositol phosphate; PKC: Protein kinase C; PLC: phospholipase C; PTEN: Phosphatase and tensin homolog, RAF: proto-oncogene serine/threonine-protein kinase; RAS: rat sarcoma; Sck: Shc-related adaptor protein; SHC: Src homology 2 domain containing; SOS: son of sevenless; SRC: Proto-oncogene tyrosine-protein kinase Src; VRAP: VEGFR-associated protein; VEGF: vascular endothelial growth factor; VEGFR2: vascular endothelial growth factor receptor-2.

## Data Availability

Datasets generated during and/or analyzed during the current study are available from the corresponding author on reasonable request.

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
