# Peer review of "Prediction of Response to Anti-Angiogenic Treatment for Advanced Colorectal Cancer Patients: From Biological Factors to Functional Imaging"

_cancers, 2024, doi:10.3390/cancers16071364_

Round 1

Reviewer 1 Report

Comments and Suggestions for Authors

This paper is a review article on colorectal cancer (CRC) occurrence and metastasis. Due to the lack of validation for predictive biomarkers, it focused on analyzing biological factors such as genetic polymorphisms, circulating biomarkers, circulating tumor cells (CTCs), circulating tumor DNA (ctDNA), and microRNA. Additionally, it highlighted the promising role of radiomics, a recently developed technique, in correlating molecular medicine with radiological phenotypes. Overall, it is deemed suitable for publication in the Cancers journal as a review article that contributes to understanding colorectal cancer.

Reviewer 2 Report

Comments and Suggestions for Authors

This manuscript is a comprehensive review article summarizing the current efforts on predicting the therapy outcome in advanced colorectal cancer patients. The authors provided wide-ranging references including case studies results for discussion, which is appreciated. One overarching suggestion is to restructure the body of this manuscript using bullet points or numerical way to flow better. Specific comments are listed below.

1.       In general, this manuscript requires some level of English editing to correct grammar errors, rephrase odd sentences and such. Also it is strongly recommended combining sentences into paragraphs as too many paragraphs just contains one sentence (e.g., Lines 135-149 and more). In some other cases, sentences are too long connected by many semicolons and those can be broken down into individual sentences for a better flow to the readers (e.g., lines 162-166 and more). Defining acronyms in the first place is needed.

2.       Several terms seem to be misused and cause the confusion. For example, biomarkers are defined as biological molecules; while in many cases in this manuscript, biomarker refers to a tool or approach (e.g., imaging or tomography). Such confusion should be corrected throughout the text.

3.       Suggested adding background information to the INTRODUCTION regarding first and second lines of treatment, including specific drug names and the brief mechanism. These information will help the readers make connection to the statements related to specific drug treatment in case studies in later sections.

4.       Section 2 - Worth specifying that VEGF-E is a viral form since it is shown in Figure 1. Also VEGF-F is a component of snake venom. Or the authors can remove VEGF-E from the figure and focus on human form.

5.       Adding clear figure legend to each figure and describe the meaning and intent of the graph with defined acronyms would be helpful.

6.       Comprehensive case studies were provided in this review article. It would be helpful to specify whether these studies were conducted specifically for colon cancer.

7.       Line 296 - clarify what "liver involvement" means.

8.       Strongly suggested making Section 3.5 as a separate sections and revising the title (e.g., advanced quantitative imaging approaches in disease detection and outcome prediction). This section seems to more focus on modern technologies that can be used to identify disease signature or patterns, not really a "biomarker" by definition. How are these imaging approaches related to predict the survival, outcome and drug potency? It would be helpful to expand a bit more on how they are related to anti-angiogenic treatment.

9.       Are the legends on page 10 supposed to be with the figures?

Comments on the Quality of English Language

see overall comments which include suggestions for English editing.

Reviewer 3 Report

Comments and Suggestions for Authors

This review concentrates on current status of the molecular bases of angiogenesis in metastatic colorectal cancer and provides an overview on potential predictive factors for treatment response to anti-angiogenic drugs in mCRC patients. This seems to be an interesting topic since colorectal cancer is a leading tumor worldwide where the the angiogenic pathway plays a crucial role in cancer spreading. The review is concise but informative supported by up-to-date literature.

Comments on the Quality of English Language

Although the text is completely understandable I would suggest English proofreading because some of the sentences are a bit awkward. Maybe engage an English lector or a native English speaker. There are also some typos. Some of the sentences that need to be rewritten are the following. Other than this, the paper is publishable.

Line 42 Great effort has been done also to correlate all these molecular markers to tumoral imaging techniques and their application to a useful tool to be used in clinical routine.

Line 51 Colorectal cancer (CRC) is one of the most common cancer worldwide accounting for

Line 85 During tumor growth, there is a balance disruption between inducers and inhibitors factors towards a pro-angiogenic signal to increase nutrient supply (Figure 1).

Line 304 thanks to the newest findings in imaging tools.

Line 331 According to the radiomics principles images are more than pictures, they are data, and such they contain a huge amount of information that can’t be analyzed only visually but need a deepest level of analysis.

Line 341 Radiogenomic uses another approach based on the newest computational tools that -should be “radiogenomics”

Line 386 Finally, one of the main problems of radiomics, which is the strong dependence on readers experience and inter-reader agreement, must be overcome in order to have more reproducible data and study types.

Line 449 Moreover, not all patients benefit from these agents and a not negligible group is thus exposed to unnecessary toxicities

Round 2

Reviewer 2 Report

Comments and Suggestions for Authors

The reviewer greatly appreciated the efforts that the authors had taken in revising the manuscript and responding to the comments. The overall quality of revised manuscript has been improved. There are still few paragraphs containing single sentence, but it's not a concern.